# CWCL: Cross-Modal Transfer with Continuously Weighted Contrastive Loss

**Rakshith S Srinivasa**     **Jaejin Cho**     **Chouchang Yang**

**Yashas Malur Saidutta**     **Ching-Hua Lee**     **Yilin Shen**     **Hongxia Jin**

Samsung Research America, Mountain View, CA

{r.srinivasa, jaejin.cho, c.yang1}@samsung.com
{ym.saidutta, chinghua.l, yilin.shen, hongxia.jin}@samsung.com

## Abstract

This paper considers contrastive training for *cross-modal zero-shot transfer* wherein a pre-trained model in one modality is used for representation learning in another domain using pairwise data. The learnt models in the latter domain can then be used for a diverse set of tasks in a zero-shot way, similar to "Contrastive Language-Image Pre-training (CLIP)" [1] and "Locked-image Tuning (LiT)" [2] that have recently gained considerable attention. Most existing works for cross-modal representation alignment (including CLIP and LiT) use the standard contrastive training objective, which employs sets of *positive* and *negative* examples to align similar and repel dissimilar training data samples. However, similarity amongst training examples has a more continuous nature, thus calling for a more *non-binary* treatment. To address this, we propose a novel loss function called Continuously Weighted Contrastive Loss (CWCL) that employs a continuous measure of similarity. With CWCL, we seek to align the embedding space of one modality with another. Owing to the continuous nature of similarity in the proposed loss function, these models outperform existing methods for zero-shot transfer across multiple models, datasets and modalities. Particularly, we consider the modality pairs of image-text and speech-text and our models achieve 5-8% (absolute) improvement over previous state-of-the-art methods in 0-shot image classification and 20-30% (absolute) improvement in 0-shot speech-to-intent classification and keyword classification.

## 1 Cross-modal alignment and transfer

Learning visual representations using natural language supervision has proven to be a powerful way to unlock impressive zero-shot performance on a number of downstream tasks [1; 2; 3; 4; 5; 6]. In this paper, we draw inspiration from these works and study the task of *cross-modal alignment for zero-shot transfer* for pairs of modalities. Let $\mathcal{U}$ and $\mathcal{V}$ denote a pair of modalities. For example, $\mathcal{U}$ may be text modality, and $\mathcal{V}$ maybe image modality. We are interested in the following: given a pre-trained model $f_\theta : \mathcal{V} \rightarrow \mathcal{Q}$ for data in $\mathcal{V}$ (where $\mathcal{Q}$ denotes the embedding space), how can we use a paired dataset of the form $\{u, v\}, u \in \mathcal{U}, v \in \mathcal{V}$, to best learn a model $g_\phi : \mathcal{U} \rightarrow \mathcal{P}$ (where $\mathcal{P}$ is the embedding space corresponding to $\mathcal{U}$) such that the learnt structure in the embedding space $\mathcal{Q}$ can be aligned with that of $\mathcal{P}$? Once trained, the models $g_\phi$ and $f_\theta$ can be used on a diverse set of downstream tasks including interfacing with Large Language Models (LLMs) (see [7]) in a zero-shot way, thus avoiding the need for costly, task-specific, labeled datasets.

Our motivation in studying the above problem lies in the fact that powerful pre-trained models existing in certain modalities, but are lacking in other modalities. For example, the recent advances

37th Conference on Neural Information Processing Systems (NeurIPS 2023).

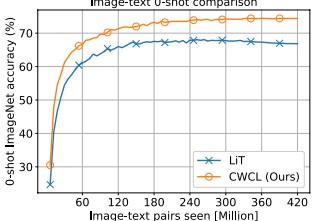 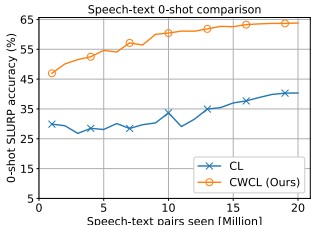

Figure 1: Comparison of zero-shot transfer performance between baseline CL and proposed CWCL. (Left): zero-shot image classification accuracy measured across training epochs for the image-text modality pair. (Right): zero-shot speech-to-intent classification measured across training epochs for the specch-text modality pair. CWCL consistently performs better than CL.

in language models have resulted in very powerful models to process text data, while no such models exist for speech and audio data. Unlike text based models that can now generalize to new tasks in a zero-shot way, speech and audio models are still trained in a task-specific way (for example, automatic speech recognition (ASR)). Further, collecting labeled datasets in speech domain offers its own set of challenges including quality control, noise, removing silence to name a few [8; 9]. Similarly, even when pre-trained models are available for certain modalities such as images, there might be challenging sub-modalities (or domains) like medical imaging on which pre-trained models may not be trained on [10]. However, large scale *paired datasets* maybe available, which connect the above modalities. For example, large datasets of speech and the associated (possibly noisy) transcripts are available easily on the internet. Similary, pairs of text and images, pairs of medical and raw text [10] maybe more easily available. Based on this observation, methods have been proposed to train image and text encoders by aligning features corresponding to paired image and text data [1; 3]. Upon training, these models demonstrate impressive zero-shot performance on a number of downstream tasks such as image classification and image-text retrieval. While in these works both encoders are trained from scratch, authors in [2] showed that using a *frozen* pre-trained image classification model as the image encoder and only training the text encoder significantly boosts downstream zero-shot performance. We observe that this abstract concept of *using a pre-trained model in one modality to supervise models in another modality using pairwise data* can then be applied to any pair of modalities.

Our main focus in this paper is on **how best to train such cross-modal models that leverage pre-trained models in one modality.** We find that standard contrastive loss used in training such models is *inefficient* at *maximizing the amount of supervision* that can be extracted from the pre-trained models. In particular, to learn the embeddings in the "unfrozen" modality, existing methods only use the embedding of the corresponding paired data from the other modality for supervision. However, there maybe many samples from the supervising modality that are similar, and to various degrees of similarity. To address this inefficiency, we propose a new loss function called **continuously weighted contrastive loss (CWCL)** for contrastive training of multi-modal models. The proposed loss function leads to better supervision and hence better alignment between the two modalities.

We study our proposed loss function using two pairs of modalities, image-text and speech-text. For image-text pair, we find that the proposed loss function leads to an **improvement of 6-8% (absolute)** compared to the best baseline on zero-shot image classification tasks. For speech-text, it leads to a **20-30% (absolute) improvement** on zero-shot speech-to-intent classification and zero-shot keyword spotting tasks. Further, our models achieve performance comparable to models trained with supervision using task-specific datasets. As shown in Figure 1, we find that models trained using the proposed loss function are data and compute-efficient. They achieve higher accuracy with fewer pairs of data samples during training. Further, embeddings of downstream test datasets generated using our models show strong alignment among data that belong to the same class, even though the models have never been exposed to these datasets. We show an example in Figure 2, where embeddings extracted from speech signals in the SLURP test dataset show significantly improved sense of similarity for data from the same class, even though no label information was provided to the model.

## 2 Continuously weighted contrastive loss

### 2.1 Existing frameworks for contrastive training

Various forms of contrastive learning has been successfully employed in both self-supervised learning [11; 12; 1; 2; 13; 14] and in supervised learning [15].
**Contrastive loss for self-supervised and multi-modal learning:** The traditional contrastive loss

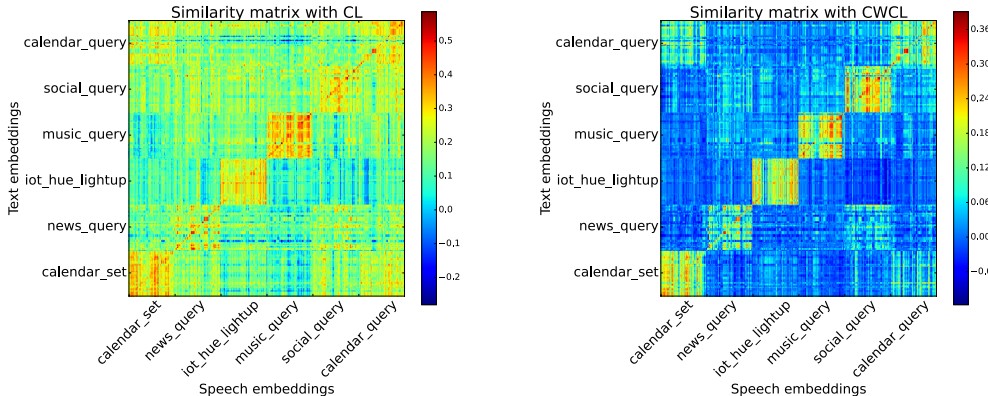

Figure 2: The similarity matrix between embeddings of the two modalities that are aligned via (Left): baseline CL and (Right): proposed CWCL. The axis labels correspond to the intent of utterances (for example, "news_query" represents utterances with news-related questions). CWCL results in a more "block" diagonal pattern than CL, indicating that speech and text samples with the same intent are more aligned while samples with different intents are more separated. This can be attributed to the continuous weighting mechanism of CWCL. Note that these embeddings are from a *downstream test dataset* which was never exposed to the model during training. The visualization confirms that CWCL leads to a higher degree of alignment between similar data samples.

function is used in both single-modality self-supervised learning as well as multi-modal alignment. We explain the formulation used in multi-modal alignment and briefly explain how the same function is used in the single-modality setting. Let $\mathcal{B}$ denote a batch of training data consisting of pairs of data samples from two modalities of size $N$: $\mathcal{B} = \{(u_i, v_i)\}_{i=1,\cdots,N}$. , where $u_i$ is from modality $\mathcal{U}$ and $v_i$ is from modality $\mathcal{V}$. Let $u_i$ and $v_i$ be encoded into embeddings denoted as $p_i$, $q_i$ respectively. This can be done by separate, modality-specific encoders or by shared encoder. Then, the traditional contrastive loss function (CL) (to align $\mathcal{U}$ with $\mathcal{V}$) is defined over $\mathcal{B}$ as

$$\mathcal{L}_{CL, \mathcal{U} \to \mathcal{V}} = \frac{-1}{N} \sum_{i=1}^{N} \log \frac{\exp\left(\langle p_i, q_i \rangle / \tau\right)}{\sum_{j \in [N]} \exp\left(\langle p_i, q_j \rangle / \tau\right)}, \tag{1}$$

where $[N]$ denotes the set $\{1, 2, \cdots, N\}$. Note that a similar loss function $\mathcal{L}_{CL, \mathcal{V} \to \mathcal{U}}$ maybe defined and the total loss function is given as $\mathcal{L}_{CL, \mathcal{U} \to \mathcal{V}} + \mathcal{L}_{CL, \mathcal{V} \to \mathcal{U}}$. By minimizing (1), the encoders *learn to align pairs of data*. Note that in doing so, for each $u_i$, $v_i$ is considered as a *positive example* and all other samples $\{v_i\}_{j \in [N], j \neq i}$ are considered to be *negative examples*. This is also illustrated in Figure 4, where the diagonal matrix indicates the set of positive examples chosen (for each row and column). As an example, in [1; 2], for each image, *only the corresponding text* is used as a positive example and *all other text samples* are used as negative examples (and vice-versa).

**Contrastive loss for supervised learning:** It is conceivable that in a given training batch, *there is more than one "positive" sample*. However the information about which samples are related to each other may be missing in self-supervised learning. However, this information is available in a supervised learning setup. Let $\mathcal{T}$ denote a batch of training data of size $M$ consisting of samples and labels: $\mathcal{T} = \{(x_i, y_i)\}$. Further, let $z_i$ be the embedding generated by the model. Then, it is clear that the set $\mathcal{P}_i = \{x_j, j \neq i | y_j = y_i\}$ forms *a set of positive examples*. This idea was explored in [15], where the following loss function [1] was proposed to leverage the label information:

$$\mathcal{L}_{\text{supcon}} = \frac{-1}{M} \sum_{i=1}^{M} \frac{1}{|P(i)|} \sum_{j \in P(i)} \log \frac{\exp\left(\langle z_i, z_j \rangle / \tau\right)}{\sum_{k \in [N], k \neq i} \exp\left(\langle z_i, z_k \rangle / \tau\right)}. \tag{2}$$

Note that the above loss function can be interpreted as *taking the average of pair-wise $\mathcal{L}_{CL}$ over the positive set*. The authors show that a combination of the above loss and the task loss yields better performance than using the task loss alone. However, this method *requires labeled datasets*.

---

[1]The authors also propose another variant of the supervised contrastive loss function. Although we do not discuss it here, it is similar in spirit to (2).

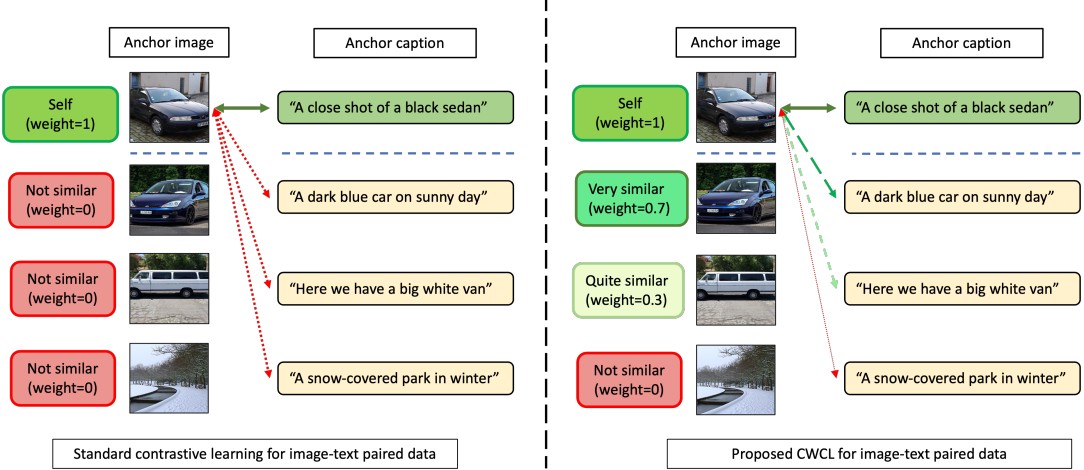

Figure 3: Existing contrastive learning methods treat samples in a batch as either strictly positive or negative. However, similarity between data samples has a more continuous and non-binary nature. In this figure, we provide an example of the nature of similarity in the context of paired image-text data. Note that the 'weight' terms in the figure are contrived for illustration purposes. The proposed CWCL loss function attracts all other data samples to a degree proportional to their similarity. Similarity itself is measured using intra-modal inner product between samples' embeddings.

In the above two loss functions and other similar variants studied in the literature, we find two shortcomings. Firstly, **other similar examples that may be present in the training batch are not considered**. In the self-supervised setting, all the other similar samples are considered as negative examples. In the supervised setting, some classes might be similar to each other (for example, multiple breeds of dogs), but are considered to be negative examples to each other. Secondly, **similarity is considered to be binary**. As a result, *all "positive examples" are attracted equally, and all "negative examples" are repelled equally.* However, we observe that *samples in a training batch maybe similar to each other to varying degrees.* Some samples might be *more similar* to each other, a few others less so many others may be *dissimilar*. For a more detailed explanation, see Figure 3.

## 2.2 Can we account for non-binary similarity?

To address the above shortcomings, we propose a novel loss function called Continuously Weighted Contrastive Loss (CWCL). We use the same setup as that in multi-modal training used to define (1). The loss function (to align $p_i$ with other $q_j$'s) is defined as

$$\mathcal{L}_{\text{CWCL}, \mathcal{U} \rightarrow \mathcal{V}} = \frac{-1}{N} \sum_{i=1}^{N} \frac{1}{\sum_{j \in [N]} w_{ij}^{\mathcal{V}}} \sum_{j \in [N]} w_{ij}^{\mathcal{V}} \cdot \log \frac{\exp(\langle p_i, q_j \rangle / \tau)}{\sum_{k \in [N]} \exp(\langle p_i, q_k \rangle / \tau)}, \tag{3}$$

where $w_{ij}^{\mathcal{V}}$'s denote the **intra-modal similarity weights** between $v_i$ and $v_j$ in modality $\mathcal{V}$. Note that a similar loss function to align modality $\mathcal{V}$ with modality $\mathcal{U}$ maybe defined, with the intra-modal similarity weights computed between $u_i$ and $u_j$. We will refer to the intra-modal similarity weights simply as weights for ease of usage and we will drop the superscript, unless the modality needs to be specified. Note that the weights are computed *pair-wise*, within each training batch.

Before we describe how these weights may be computed, we highlight the properties that they need to have. Firstly, we normalize the weights to be between 0 and 1: $w_{ij} \in [0, 1]$. Secondly, *"similar" samples from within a given domain should have higher weights and dissimlar samples should have lower weights.* With these properties, note that $\mathcal{L}_{\text{CWCL}}$ provides a way to interpolate between the self-supervised and fully-supervised variants described earlier. When the weights are given as $w_{ij} = \mathbb{1}_{\{i\}}(j)$ where $\mathbb{1}_{\mathcal{S}}$ denotes the indicator function w.r.t set $\mathcal{S}$, it is equivalent to $\mathcal{L}_{\mathcal{CL}}$. On the other hand, in the supervised setting, if $w_{ij}$ is defined as $w_{ij} = 1$ for all pairs $i, j$ belonging to the same class, but 0 otherwise, it is equivalent to $\mathcal{L}_{\text{supcon}}$. More importantly, $\mathcal{L}_{\text{CWCL}}$ allows the model to use a **a continuous sense of similarity**, by i) computing a softmax function for all pairs in the

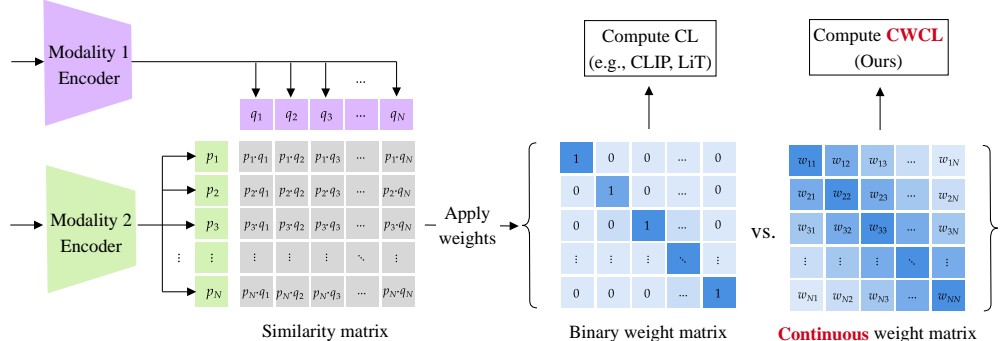

Figure 4: The classical CL-based methods (e.g., CLIP [1], LiT [2], etc) can be interpretes as using a binary weight matrix for choosing the positive examples. The proposed CWCL utilizes a continuous weight matrix to account for the non-binary nature of similarity for improved alignment.

training batch (inner summation in Equation. (3)) and ii) weighting these softmax terms by the similarity weights. Further, note that all pair-wise inner products are already computed even in (1), (2). Therefore, computing $\mathcal{L}_{\text{CWCL}}$ is similar in computational complexity to $\mathcal{L}_{\text{CL}}$ and $\mathcal{L}_{\text{supcon}}$.

## 2.3   How can we obtain intra-modal similarity weights?

In the traditional self-supervised setting, no information about similarity between training data points maybe available. This might also be the case in multi-modal learning such as in[1], where the modality encoders are initialized randomly. However, authors in [2] explored the idea of using pre-trained models as initialization in multi-modal models. Further, they find that *freezing* the pre-trained model (except maybe for a final linear layer) yields the best performance. This setup offers a natural way to obtain the similarity weights. We can measure the similarity between the embeddings from the pre-trained model. We focus on this setup, where we use frozen, pre-trained models for one modality to train models in another modality. Note that even though the model encoding the first modality is frozen, follow-up layers maybe added and trained. Let $\mathcal{V}$ be the "frozen" modality with a pre-trained initialization. Then to align modality $\mathcal{U}$ with $\mathcal{V}$ using (3), $w_{ij}^{\mathcal{V}}$ maybe computed as $w_{ij}^{\mathcal{V}} = \langle q_i, q_j \rangle / 2 + 0.5$ in order for $w_{ij} \in [0, 1]$. We do not explore other formulations in this paper. A natural question is about how such weights can be computed for the modality $\mathcal{U}$. If the model in modality $\mathcal{U}$ is also initialized using a pre-trained model, the weights may be computed in a similar way. However, in this paper, we only focus on the *cross-modal transfer*, with similarity weights being computed only in the *frozen modality* initialized with a pre-trained model. Assuming the modality $\mathcal{V}$ is the frozen one, our loss function is given as

$$\mathcal{L}_{\text{cross-modal transfer}} = \mathcal{L}_{\text{CWCL}, \, \mathcal{U} \to \mathcal{V}} + \mathcal{L}_{CL, \, \mathcal{V} \to \mathcal{U}}. \tag{4}$$

Note that the exact configuration and choice of which modality to freeze will depend on the pairs of modalities being considered, the quality of pre-trained models and paired datasets available.

## 3   Related work

**CLIP like models:** CLIP [1] and ALIGN [3] introduced a set of foundational vision-language models where the encoders, one for the image, another for the text modality output embeddings in a shared space. In this set of works the encoders for all the modalities are randomly initialized and trained from scratch. Other works have looked at extending the concept to other modalities like image-audio [16], others have explored richer embedding [17], adaptive prompt learning [5], architectural advancements like Mixture-of-Experts [6]. Some works also considered the problem setting where embeddings of both image and text are processed together so that specialized text query relevant image embeddings can be obtained [18; 19]. Another notable work, [4], the authors obtained impressive performance by improving individual encoders by processing image and text embeddings together to minimize caption-loss. Additionally, [20] proposed an extension to the cross-modal contrastive loss that can leverage labeled training data by combining it with SupCon [15]. Recent works such as [21; 22; 23] consider the alignment between images and text that do not belong to the same pair, similar to our proposed method. In [21; 23], both encoders are trained from scratch

by using a self-distillation process. Such a training process requires careful parameter tuning and generally has lower performance ([21] achieves about 42.4% 0-shot on ImageNet) compared to using pre-trained models, as demonstrated by the metrics. Another difference between our work and the above works is that we consider intra-modal similarity to attract image-text pairs. Owing to the availability of strong pre-trained uni-modal models, intra-modal offers a clear way to identify similar data samples. In [22], the authors consider using a third, object detection model to obtain similarity between images. However, their method is specific to image-text modality pair. It may also lead to performance degradation, as seen in the zero-shot metrics reported.

**LiT like models:** Alternatively, LiT [2] proposed the idea of leveraging strong pretrained models in one domain and aligning the embeddings of another domain to the pretrained model's embedding space. Works in this line have looked at extending to multiple domains like image-audio-text [24], music-audio [25], speech-text for speech translation [26]; fine-grained query specific image embeddings [27] and benefits of cross-modal alignment for regularizing unimodal classifiers [28]. Along with building a model capable of conversation, [29] proposed the use of cross-modal attention layers to improve image-text cross-modal alignment. However, none of these works consider the problem of similarity across samples within the same modality that is explored in our work. Further, all these works are complementary to CWCL and can be improved by it. Handful of works explore similarity amongst samples [30; 13; 30] propose removing certain samples from the negative set used to compute contrastive loss (1) if their average similarity to the other samples in the batch is greater than a certain threshold [30]; [13] propose using a threshold on similarity to decide which samples are positive pairs and negative pairs and combining it with [15]. However, the above works still consider similarity as a binary entity, where as CWCL uses a continuous model.

**Incorrect negatives in contrastive learning:** Contrastive learning incorrectly assumes that for a given sample, every other sample in the dataset is dissimilar [31]. In the self-supervised learning one of the remedies proposed is to re-weight the negative part of the contrastive loss' denominator to account for the presence of similar (or positive) samples [32]. However, in the case of cross-modal alignment with pretrained models, the pretrained model is a better indicator of the similarity [30; 13].

# 4  Experiments

In this section, we provide experimental results that demonstrate that CWCL leads to better zero-shot transfer performance. We study two pairs of domains, namely image-text and speech-text. For image-text pair, we demonstrate zero-shot transfer to image classification and image/ text retrieval. On both tasks, CWCL shows improved performance over existing methods for zero-shot transfer. Next, we report results for speech-text modality pair, where we consider the tasks of speech-to-intent classification and keyword spotting. Given the difficulties in collecting task-specific speech datasets, we expect CWCL-based zero-shot transfer to have a large impact in this domain. Note that our main goal is to study the effect of using CWCL. *We use open source, publicly available and easily accessible datasets for our study and leave the task of training with larger datasets to future work.*

## 4.1  Cross-modal transfer between image and text modalities

**Model architecture:** Our model architecture follows that in [2] and has a vision encoder and a text encoder. For the vision encoder, we use the ViT-L/16 model architecture [33] pre-trained on ImageNet. We compute the similarity weights using the embeddings from before the final linear layer that is not frozen during training. For the text encoder, we consider two architectures: transformer encoder architecture with 12 layers,output dimension 768, and number of heads set to 12 and we also consider the BERT-large architecture.

**Datasets for contrastive training:** All our experiments are based on the combination of two publicly available datasets, CC12M and YFCC15M. The CC12M dataset is a subset of the Conceptual Captions dataset [34] defined in [35]. We use a set of 10 million images that are still available in the set of URLs (since the rest of them have been taken down). The YFCC15M dataset is a subset of the Yahoo Flicker Creative Commons dataset [36] defined by [1] by filtering for high quality English text. It contains a set of 15 million image-text pairs. Model training details are provided in A.1.

### 4.1.1  Zero-shot image classification

For zero-shot image classification, we experiment on 5 datasets: ImageNet [37] validation, ImageNet-V2 [38], ImageNet-R [39; 40], ImageNet-A[41]and ObjNet [42], similar to [2]. We provide our

experimental results in Tables. 1, 2. The results for SimCon [13], and LiT [2] are obtained from our own experimentation. For [13], we use their loss function in our set up. For [2], we use their recommended experimental settings from their paper. Note that the metrics are obtained by using the same model architecture and dataset for all the methods being compared. CWCL yields a significant boost over the other methods in zero-shot performance. Further, as shown in Figure1, CWCL achieves higher accuracy with fewer image-text training pairs. Owing to the CWCL formulation, the text embeddings generated by our model are designed to be similar to a larger set of similar images than the baseline methods, hence leading to better generalization.

Table 1: Zero-shot image classification performance using the ViT-L/16 + 12-layer transformer configuration. **CWCL achieves a significant improvement in zero-shot image classification** across multiple datasets, including out-of-domain datasets such as ObjectNet.

| Method | ImageNet (%) | ImageNet-V2(%) | ImageNet-R(%) | ImageNet-A(%) | ObjNet(%) |
|---|---|---|---|---|---|
| CLIP | 31.3 | - | - | - | - |
| OpenCLIP | 34.8 | 30 | - | - | - |
| SimCon | 67.9 | 58.57 | 59.32 | 37.16 | 44.9 |
| LiT | 66.84 | 58.82 | 61.28 | 37.31 | 45.08 |
| **CWCL (Ours)** | **74.41** | **66.25** | **67.37** | **45.58** | **50.5** |

Table 2: Zero-shot image classification using the ViT-L/16 +BERT-large configuration . CWCL-based training achieves **state-of-the-art** (when trained on publicly available datasets) performance on all of zero-shot experiments.

| Method | ImageNet (%) | ImageNet-V2(%) | ImageNet-R(%) | ImageNet-A(%) | ObjNet(%) |
|---|---|---|---|---|---|
| LiT | 71.2 | 62.98 | 63.8 | 40.28 | 48.1 |
| **CWCL (Ours)** | **76.48** | **67.86** | **68.7** | **47.27** | **52.38** |

### 4.1.2 Zero-shot Image-text retrieval

We also examine the zero-shot image-text retrieval capabilities of our proposed method. Note that our experiments are only towards comparing standard contrastive loss with CWCL. We leave the task of training with larger datasets [1; 2; 3] and using multi-objective training (which maybe used along with contrastive tuning to obtain better retrieval performance) [34; 29; 19] for future exploration. In our experiment, we simply compare the performance of models trained with contrastive loss (as done in [2]) to that of models trained using CWCL. We use the MS-COCO validation dataset [43] to study zero-shot retrieval performance of these models. We report our results in Table 3. Retrieval metrics for the ViT-L/16+12 layer transformer configuration model are provided in Table 6 in the Appendix. Models trained with CWCL outperform those trained using the standard contrastive loss function.

Table 3: Zero-shot retrieval results on MS-COCO dataset by using ViT-L/16+BERT-large configuration as the image and text encoders respectively.

| Method | I →T retrieval | | | T→I retrieval | | |
|---|---|---|---|---|---|---|
| | R@1 | R@5 | R@10 | R@1 | R@5 | R@10 |
| LiT | 34.58 | 59.78 | 70.68 | 28.49 | 54.04 | 65.87 |
| **CWCL (Ours)** | **40.36** | **66.62** | **77.76** | **30.04** | **54.84** | **66.06** |

### 4.1.3 Robustness to templates for zero-shot classification

An added benefit of the proposed CWCL formulation is that our model is robust to the templates/ prompts used in zero-shot tasks. In zero-shot image classification, the labels are converted to text prompts in order to adapt the task of classification to that of alignment. In particular, both [1; 2] use a set of 80 "template" sentences to convert each label into 80 sentences, extract the text embeddings for all the sentences and use their mean embedding as the representation of the corresponding class. We expect that CWCL leads to robustness w.r.t the choice of such templates or prompts. We study this

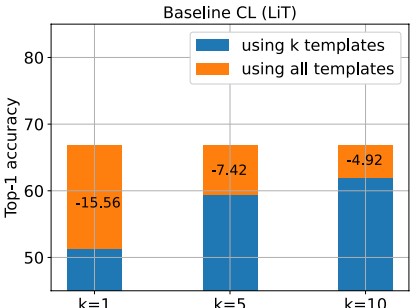 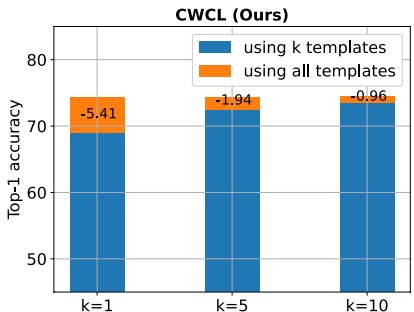

Figure 5: Comparison of robustness to templates. (Left): the baseline CL method of LiT [2]. (Right): the proposed CWCL approach. The number displayed at each bar reflects the decrease in accuracy due to using only a subset of templates compared to using the full set.

by changing the number of template sentences used to build the classifier embeddings. In particular, we design simple templates such as "this is a photo of ", and "this is an image of " and experiment over $k = 1, 5, 10$ templates. We provide further details on the templates in Section A.1.2. We report the results for our model and that of [2] in Figure 5. As can be seen, models trained using CWCL are able to obtain peak performance with fewer number of templates, whereas models trained using standard contrastive loss require a higher number of templates to build better classifier embeddings. We believe that robustness to the choice and the number of template sentences/prompts used is crucial to improve the ease of usage of such models.

## 4.2 Cross-modal transfer between speech and text modalities

The proposed method can be applied to speech-text cross-modal learning to transfer semantic knowledge from text embeddings to speech embeddings. Speech models with language understanding are desired for tasks in the field of spoken language understanding (SLU) [44; 45]. SLU differs from automatic speech recognition (ASR), which simply generates a transcription, but does not have language understanding. SLU models, unlike ASR models can then be used on a wide variety of downstream tasks such as intent classification (in multiple domains) [8], keyword spotting [46].

In general, speech pre-training schemes usually include information about the phonemes or paralinguistic information (e.g. speaker, emotion, pathology, etc.), but they do not include semantics in language. While some works have explored the usage of contrastive learning to train SLU models, they use the standard contrastive training method [47]. However, similar to the image-text case, this may not be efficient. For instance, "Turn on the volume", is closer in meaning to "Increase the sound" than "Set an alarm for 7:00 AM tomorrow morning". In this case, the standard cross-modal contrastive loss is unable to learn the cross-modal relationship between the text of the first sentence and the speech of the second sentence since they are considered to be a "negative pair". This is precisely what is address by CWCL. As we demonstrate later, CWCL achieves a significant boost in performance on downstream tasks.

We train a speech-text multi-modal model with a dataset where speech and its corresponding transcript are available. Note that this is a generic dataset that is not specific to any SLU task. We use a pre-trained, frozen text encoder, owing to the availability of strong pre-trained language models. A trainable linear layer is added on top of the frozen text encoder to match the dimensionality of the speech and text embedding. We also use a pre-trained speech encoder that is robust to diverse acoustic condition and further train it using the proposed loss function.

**Model architecture:** For the speech model, we used the encoder of the pre-trained Whisper ASR [48], which is expected to be robust to different acoustic conditions. For the text models, we found 49 publicly available hugging face models by searching with filters as, task: Zero-shot classification, libraries: Transformers, and languages: English. We manually added one RoBERTa-based model fine-tuned on MASSIVE [49] data. All 50 models were compared on zero-shot text-to-intent classification using the SLURP dataset [8] and the top 2 models were selected. The best model (we call it RoBERTa+S) was the RoBERTa-based model fine-tuned on MASSIVE data since the data includes SLURP (only the text data) [2]. The second model was a BART-based model fine-tuned on Yahoo Answers topic classification (we call it BART+Y) [3].

---

[2] https://huggingface.co/qanastek/XLMRoberta-Alexa-Intents-Classification
[3] https://huggingface.co/joeddav/bart-large-mnli-yahoo-answers

**Datasets for contrastive training** For cross-modal training, we used the Common Voice Corpus 13.0 [50]. This dataset consists of roughly 2400 hours of speech data and the corresponding transcripts obtained using crowd-sourcing and includes speech from a diverse set of demographics across age and gender. We use the English subset. Model training details are provided in A.2.

### 4.2.1  Zero-shot speech-to-intent classification

After the cross-modal embedding alignment stage, we evaluated the models on the zero-shot speech-to-intent classification task. The task is to classify a given speech sequence into one of the intent classes. The main difference between the zero-shot and supervised intent classification is the zero-shot classification can be done without training a classifier.

**Class embedding generation:** Similar to the image-text case, we compute the embeddings of a given speech signal and compute its similarity with the text embeddings for all the intent classes. These class embeddings are obtained as averaged embedding of text sentences' embeddings of the corresponding classes. During inference, the class embedding that has the highest similarity score with the input speech embedding is chosen as the predicted class.

**Dataset:** We used the SLURP [8] and STOP [51] datasets for evaluation. In the SLURP dataset, we used all the text sentences in the *train* subset to generate the class embeddings for 60 intent classes where intent is defined as the concatenation of scenario and action labels, following ESPnet-SLU [52]. We did not use the *train_synthetic* subset since more than half of the text sentences overlap with the *devel* and *test* subsets. On average, 191 text sentences were used per class. We compare the systems by evaluating them on the *devel* and *test* subsets. In the STOP dataset, we used the 8 unique domain labels as intent labels. Although not intents in a strict sense, the domain labels can be considered a simpler version of the intent labels. Since significantly more sentences are available in STOP, we randomly extracted 200 sentences per domain from the training set to generate the class embeddings. The evaluation was done on the validation and test sets.

**Results**: In previous works, speech-to-intent classification has been done with an ASR-NLU pipeline system where the speech is first transcribed by ASR (speech-to-text), after which the transcription is classified into an intent using NLU (text-to-intent) [8]. We refer to the text-to-intent performance achieved by the pre-trained text encoders as the "reference" performance. This provides an estimate of the performance that can be expected from the speech encoder on the speech-to-intent task.

The first results of speech-to-intent classification are shown in the SLURP and STOP (without the superscript [#]) columns in Table 4. In all cases, multi-modal training with the CWCL loss outperformed the CL loss. On the SLURP dataset, RoBERTa+S has a higher reference performance compared to BART+Y because the fine-tuning data for RoBERTa+S included the SLURP text data. This also leads to a better performance compared to using the BART+Y model as the text encoder.

On the STOP dataet, RoBERTa+S has a lower reference compared to BART+Y, implying that the RoBERTa+S' text model overfits the SLURP data. However, the RoBERTa+S-based speech intent classification was still better than the BART+Y-based one. This implies that the text model architecture could be another factor that contributes to transferring performance to the speech model. To be specific, the RoBERTa+S was RoBERTa which consists of only encoder layers while the BART+Y was the encoder-decoder-based BART model. Another thing to note is that CWCL with RoBERTa+S outperforms the text-to-intent reference performance on the STOP (87.87 vs. 84.78) dataset. This is because, during the cross-modal alignment stage using CWCL, the speech tower might have learned how to utilize acoustic cues in addition to linguistic information from a given speech utterance, to align its embedding to the semantic embedding from the text tower. However, this did not happen in the case of SLURP, because the SLURP dataset includes more intent classes than STOP (60 vs 8 classes), thus being more challenging in transferring knowledge from text to speech during the cross-modal alignment stage.

**Experimenting with different templates to generate text embeddings:** So far, each class embedding used in zero-shot intent classification was generated by averaging all the corresponding text sentences' embeddings from the class in the training subset. Although collecting the text data with the intent labels can be less expensive than collecting speech data with the intent labels, the former may not always be possible. To address this, we manually devised a fixed set of general templates that were applied to every class. For example, templates are of the form "This audio is about [class]", and "The utterance is related to [class]", and the text embeddings are averaged to obtain the class embedding. For the exact templates we used, readers may refer to Appendix A.2.5. The results are

Table 4: Top-1 accuracy for zero-shot speech-to-intent classification (SLURP and STOP) and keyword spotting (GSCV2) after thick vertical line. Superscript $^{\#}$ is used to indicate use of general templates for class embedding extraction. Supervised results are provided in gray after the double-horizontal line: [52] is for speech-to-intent and [53; 54; 55] are for keyword spotting.

| Method | Text model | SLURP | SLURP$^{\#}$ | STOP | STOP$^{\#}$ | GSCV2 | GSCV2$^{\#}$ |
|---|---|---|---|---|---|---|---|
| CL | RoBERTa+S | 40.35 | 23.68 | 70.13 | 50.56 | 64.74 | 59.65 |
| (baseline) | BART+Y | 22.73 | 8.06 | 55.67 | 42.07 | 56.33 | 45.54 |
| **CWCL** | RoBERTa+S | 63.80 | 40.75 | 87.87 | 67.77 | 81.02 | 82.77 |
| **(Ours)** | BART+Y | 53.12 | 30.51 | 80.99 | 73.08 | 88.81 | 89.43 |
| Text-to-intent | RoBERTa+S | 88.19 | 59.86 | 84.78 | 69.10 | 100 | 98.20 |
| (reference) | BART+Y | 77.03 | 45.93 | 92.93 | 79.11 | 100 | 100 |
| ESPnet [52] | - | 77.00 | - | - | - | - | - |
| Att. RNN [53] | - | - | - | - | - | 93.9 | - |
| Wav2Vec2 [54] | - | - | - | - | - | 96.6 | - |
| M2D [55] | - | - | - | - | - | 98.5 | - |

shown in the SLURP$^{\#}$ and STOP$^{\#}$ columns in Table 4. We again observe that the proposed CWCL loss outperforms the CL loss.

**Comparison to supervised training**: We also present results of a supervised SLU model on SLURP, based on ESPnet-SLU [52]. Considering our system is zero-shot, the result is noteworthy. For STOP, we could not find previous supervised works that evaluated systems the same way.

Due to lack of space, we present the following results in A.2.2. In Table 7, we found that leveraging pre-trained models was more beneficial than training from scratch for speech-text embedding alignment. As seen in Table 8, locking the text encoder and fine-tuning the speech encoder gave the best performance. We found that batch size is not a critical factor, as shown in Table 9.

### 4.2.2 Zero-shot keyword spotting (KWS)

We also tested our model for KWS using the Google Speech Command Dataset V2 (GSCV2) [9] where we classified among the 35 keywords in the Google Speech Command. The result is shown in the columns after the thick vertical line in Table 4. For the results in the first column (without $^{\#}$), we used each keyword as is to extract the class embedding from the text model. For the second column (with $^{\#}$), we used the general template used in the speech-to-intent experiments. The results show that the proposed method outperforms the baseline. With CWCL, the KWS$^{\#}$ outperformed KWS. This could be because the text models that generate the class embeddings are usually trained with sentence-level samples, not word-level ones whereas the keywords are words, i.e., the KWS class embeddings are extracted from words whereas KWS$^{\#}$ are extracted from sentences constructed using templates, thus resulting in better class embedding.

**Comparison to supervised training**: Interestingly, results achieved with the BART-based text model are comparable to the supervised learning mechanisms of [53; 54; 55]. Note that the self-supervised mechanisms use training data to train the final linear classifier [54; 55]. However, our models without any training data still achieve close to $90\%$ accuracy. This will be useful when defining new keywords as collecting large datasets for keyword classification becomes difficult [9]. Additional results are provided in Table 10 and in Table 11, respectively in Appendix A.2.

## 5 Conclusion

In this paper, we make the observation that existing contrastive learning based methods for cross-modal alignment using pre-trained models are not efficient in extracting supervision from the pre-trained embeddings. In particular, many similar examples that do not form pairs in the training data are ignored. We address this by developing a novel loss function that accounts for the continuous nature of similarity and uses information from all similar examples in a training batch. We train models for two pairs of modalities using this loss function, namely image-text and speech-text. In both cases, we observe a significant increase in zero-shot performance on downstream tasks. We believe that the proposed loss function will be impactful in leveraging powerful pre-trained models and transfering the learnt knowledge to other modalities and domains.

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

# A  Appendix A

## A.1  Experimental details for aligning image and text modalities

### A.1.1  Model training details

We build upon the code repository in [56]. We train our models for a total of 70 epochs, where each epoch uses a subset of 6 million images. The batch size is set to 16000. Note the number of training steps in this case is equal to 26,250. We train on 4 A100 GPUs. Note that we experimented with different sizes for the subset used in each epoch (ranging from 2 million to the full dataset) and we obtained the best performance when the size was 6 million (for our method and the baseline methods that we train). We use a learning rate of 0.001, AdamW optimizer with $\beta_1 = 0.9, \beta_2 = 0.999$ and a weight decay of 0.0001 [57].

### A.1.2  Simple templates to test model robustness

One of the advantages of cross-modal 0-shot transfer is the ability of the trained models to be used on downstream tasks without any further training. However, the downstream task still needs to be adapted to the task of modality alignment. We discuss this adaptation in the context of image classification and provide details about our experiments reported in Section 4.1.3.

In [1; 2], the downstream task of image classification task is solved by first changing the class labels to sentences. The sentences are then converted to embeddings using the text encoder. Given a test image, the text embedding that it aligns the most with determines its class. In particular, both works use a set of 80 "template sentences" to convert each label to 80 sentences. The text embedding representing a given label is then computed as the average of the embeddings of these 80 sentences.

We observe that the classification accuracy depends on the choice of these template sentences, as also seen in [5]. To illustrate this, we formulate $k = 1, 5, 10$ **simple** template sentences and use them to generate the classifier embeddings. We list these sentence in Table 5. Note that for $k = 1$, we use the first sentence only and for $k = 5$, we use the first 5 sentences. Our motivation in choosing simple sentences is to mimic the process of an end user who may not have the resources to carefully design the template sentences. Our goal is to test our model's robustenss under such a scenario. As shown in Figure 5, a model trained using standard contrastive tuning shows poor performance as the number of template sentences is reduced. This shows that to achieve high accuracy, an end user must design template sentences that are complex enough. However, a model trained using CWCL maintains its performance across varying number of template sentences, even when only simple templates are used. Our hypothesis is that owing to the continuous nature of the similarity used during training, the model has learnt better cross-modal associations.

Table 5: Simple template sentences that we use to generate classifier embeddings.

| |
| --- |
| a photo of a { } |
| an image of a { } |
| a picture of a { } |
| this is a { } |
| a snap of { } |
| a shot of { } |
| an illustration of { } |
| an example of { } |
| a { } is pictured here |
| In this picture, we can see a { } |

### A.1.3  Cross-modal retrieval

We also examine the zero-shot image-text retrieval capabilities of our proposed method. Note that our experiments are only towards comparing standard contrastive loss with CWCL. We leave the task of training with larger datasets [1; 2; 3] and using multi-objective training (which maybe used in conjuntion with contrastive tuning to obtain better retrieval performance) [34; 29; 19] for future exploration.

Table 6: CWCL improves upon the CL-based alignment method for image-text retrieval.

| Method | I→T retrieval | | | T→I retrieval | | |
|---|---|---|---|---|---|---|
| | R@1 | R@5 | R@10 | R@1 | R@5 | R@10 |
| CL | 30.42 | 54.32 | 65.82 | 24.17 | 49.04 | 61.05 |
| **CWCL (Ours)** | **35.10** | **61.52** | **73** | **25.69** | **50.04** | **61.59** |

In our experiment, we simply compare the performance of models trained with contrastive loss (as done in [2]) to that of models trained using CWCL. We use the MS-COCO validation dataset [43] to study zero-shot retrieval performance of these models. We report our results in Table 6. Models trained with CWCL outperform those trained using the standard contrastive loss function.

## A.2 Experimental details for aligning speech and text modalities

In this section, we provide additional details about training the speech-text alignemnt models.

### A.2.1 Model training details

We train each model for a total of 20 epochs, where one epoch consumes the whole training data equal to 1,013,630 samples. We use a batch size of 20 with the 12,500 warmup steps and train on 1 A100 GPU. We use a learning rate of 0.00003, AdamW optimizer with $\beta_1 = 0.9, \beta_2 = 0.999$, a weight decay of $0.0001$, and gradient clipping norm of 10.

### A.2.2 Effects of using pre-trained model weights, locking location, and batch size

Each reported number in this section is Top-1 accuracy (%) on SLURP data for speech intent classification.

**Pre-trained speech encoder vs randomly initialized**: In Table 7, we compared starting multi-modal training from scratch and from pre-trained weights. The performance is significantly boosted by initializing the speech encoder using weights from the encoder part of the Whisper ASR model [48]. However, regardless of using random weights and pre-trained model weights, training with CWCL results in a much better downstream performance.

Table 7: Comparison between using randomly initialized weights and pre-trained weights for speech encoders during training: Top-1 accuracy (%) on SLURP data

| Method | Random initialization | Pre-trained weights |
|---|---|---|
| CL | 13.80 | 22.73 |
| CWCL | **26.17** | **53.12** |

*Locking location*: We have 4 ways to lock our model during multi-modal training since we have pre-trained speech and text models. We compared all the locking options and the result is shown in Table 8. In both baseline and CWCL losses, locking the text model works best. This can be seen as transferring the knowledge of semantic relationships in text models to speech models.

Table 8: Locking location vs. performance: Top-1 accuracy (%) on SLURP data

| Locking location | none | speech | text | both |
|---|---|---|---|---|
| CL | **18.77** | 7.89 | 24.03 | 9.82 |
| CWCL | 17.50 | **27.39** | **53.12** | **16.70** |

*Batch size vs. performance*: Since the large batch size was shown to improve performance with contrastive loss in computer vision, we also did a similar experiment to see how the batch size affects the performance as it gets larger. As the batch size increases, we also increased the learning rate proportionally, e.g., if bs=20 has lr=1, bs=40 has lr=2. The results are in Table 9.

Table 9: Effective batch size vs. performance: Top-1 accuracy (%) on SLURP data

| batch size | 20 | 40 | 80 |
|---|---|---|---|
| CL | 24.03 | 25.20 | 24.51 |
| CWCL | **53.12** | **53.94** | **51.80** |

### A.2.3 Further evidence of modality alignment due to CWCL

In Figure 2, we showed the alignment (measured as inner product) between speech features and text features obtained from models trained using just CL and those trained using CWCL. We use the speech and text data from the SLURP test dataset. We illustrated that speech and text embeddings that belong to the same intent class were much more aligned compared to speech and text from mismatched classes. In this section, we provide more examples that support this observation.

In Figures 6, 7, we show the alignment between the speech and text embeddings where the speech and text samples belong to classes other than those used in Figure 2. We again see that the alignment between samples in the same class is much higher than that between samples in different classes. In general, we observe the same pattern to hold across all the classes in the dataset, thus confirming that our results are not due to sampling bias.

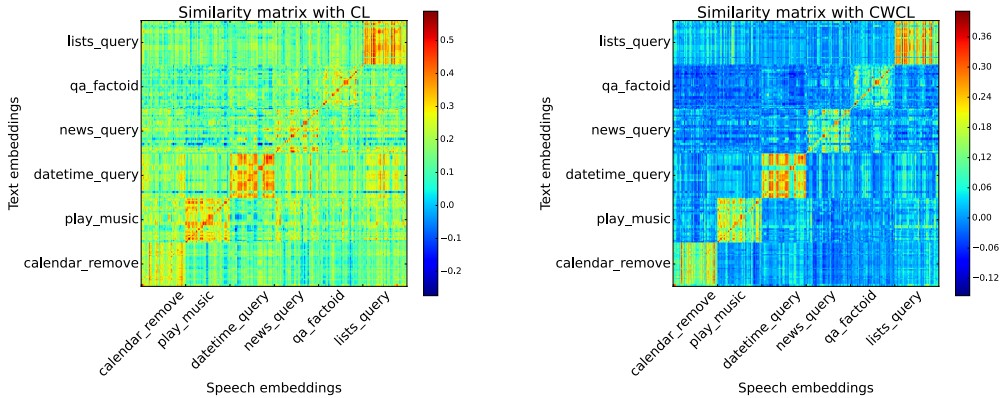

Figure 6: Cosine-similarity between speech and text embeddings obtained by sampling 6 classes randomly from the SLURP test dataset. In this case, the sampled classes are different from those used in Figures 2 and 7.

### A.2.4 Additional tables for reference

Table 10 additionally shows the Top-5 accuracy over speech-text experiments. Since most of the previous works did not report this metric, we only include our own experimental results. Table 11 shows the existing supervised model performances where the models are either trained or fine-tuned on the labeled Google Speech Command Dataset V2 for performing the keyword spotting (KWS) task, while our methods did not require any labeled KWS data for performing the task.

### A.2.5 General template as a python list

To test the speech-text alignment models, we use a "general" set of templates in addition to the one obtained by using the text from the training data itself. This general set of templates aims to mimic a scenario where the no example texts maybe available. We list the set of template sentences used in the general set here.

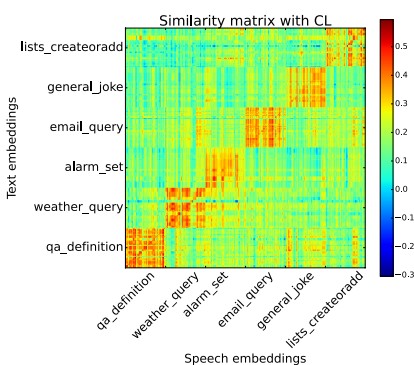
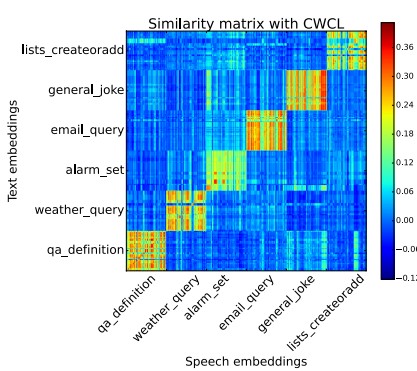

Figure 7: Another example of alignment between speech and text embeddings. The sampled classes are different from those used in Figures 2 and 6.

Table 10: Top-5 accuracy for zero-shot speech-to-intent classification (SLURP and STOP) and KWS on Google Speech Command Dataset V2. Superscript $^{\#}$ is used to indicate use of general templates.

| Method | Text model | SLURP | SLURP$^{\#}$ | STOP | STOP$^{\#}$ | KWS | KWS$^{\#}$ |
|---|---|---|---|---|---|---|---|
| CL | RoBERTa+S | 69.57 | 49.86 | 98.19 | 94.03 | 82.22 | 82.53 |
| CL | BART+Y | 52.97 | 24.87 | 95.27 | 81.63 | 84.02 | 78.14 |
| **CWCL (Ours)** | RoBERTa+S | 84.53 | 68.58 | 99.38 | 96.52 | 91.20 | 92.42 |
| **CWCL (Ours)** | BART+Y | 79.48 | 57.34 | 99.48 | 97.71 | 93.79 | 94.30 |
| Text-intent | RoBERTa+S | 95.66 | 83.36 | 98.93 | 95.20 | 100 | 98.20 |
| (upper bound) | BART+Y | 99.58 | 73.82 | 99.45 | 98.40 | 100 | 100 |

General template sentences: [ it is about { }, it was about { }, it will be about { }, this is about { }, this was about { }, this will be about { }, it is related to { }, it was related to { }, it will be related to { }, this is related to { }, this was related to { }, this will be related to { }, it is talking about { }, it was talking about { }, it will be talking about { }, this is talking about { } this was talking about { }, this will be talking about { }, I am talking about { }, I was talking about { }, I will be talking about { }, You are talking about { }, You were talking about { }, You will be talking about { }, They are talking about { }, They were talking about { }, They will be talking about { }, We are talking about { }, We were talking about { }, We will be talking about { }, it talks about { }, it talked about { }, it will talk about { }, this talks about { }, this talked about { }, this will talk about { }, I talk about { }, I talked about { }, I will talk about { }, You talk about { }, You talked about { }, You will talk about { }, They talk about { }, They talked about { }, They will talk about { }, We talk about { }, We talked about { }, We will talk about { } ]

Table 11: Keyword spotting Top-1 accuracies on GSCV2 from existing supervised models.

| Method | KWS |
|---|---|
| Attention RNN [53] | 93.9 |
| KWT-2 [46] | 97.74 |
| Wav2Vec2 [54] | 96.6 |
| M2D [55] | 95.4 |
| M2D - Fine tuned [55] | 98.5 |

