# A   Appendix A

## A.1   Detailed explanation of continuous nature of similarity

In this section, we expand on our observation that similarity between training samples is not binary. Consider the images shown in Figure 6. Let the anchor image and the four images at the bottom be part of a batch of training data (possibly along with many other samples). Note that the similarity of the anchor image ranges from 'very similar' to 'highly dissimilar' and that it is not simply binary. However, Existing methods for contrastive training only use a binary notion for similarity, and categorize the samples in a batch into "positive" and "negative" sets. As a consequence, the models fail to correctly learn associations between different data samples.

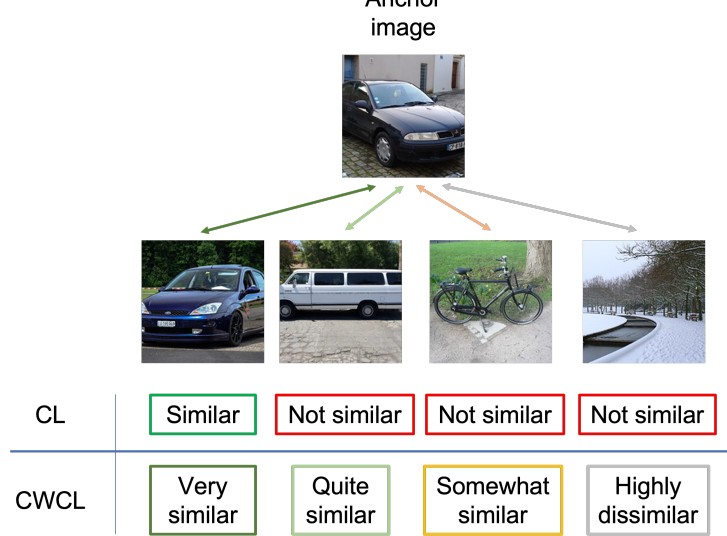

Figure 6: An illustration of the continuous nature of similarity in training data. In this example, the anchor image is similar (or dissimilar) to the other images to various degrees. Existing methods choose a subset of images and consider them to be 'positive examples', and consider the rest of the examples as 'negative' examples. Once these subsets are chosen, the embedding of the anchor image is *aligned* (to an equal degree) to those of the positive examples and *contrasted* with those of the negative examples. As a consequence, any similarity between the anchor image and the so-called 'negative' examples is completely ignored. Further, all 'positive' examples are considered to be *equally similar*, although this might not be the case.

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

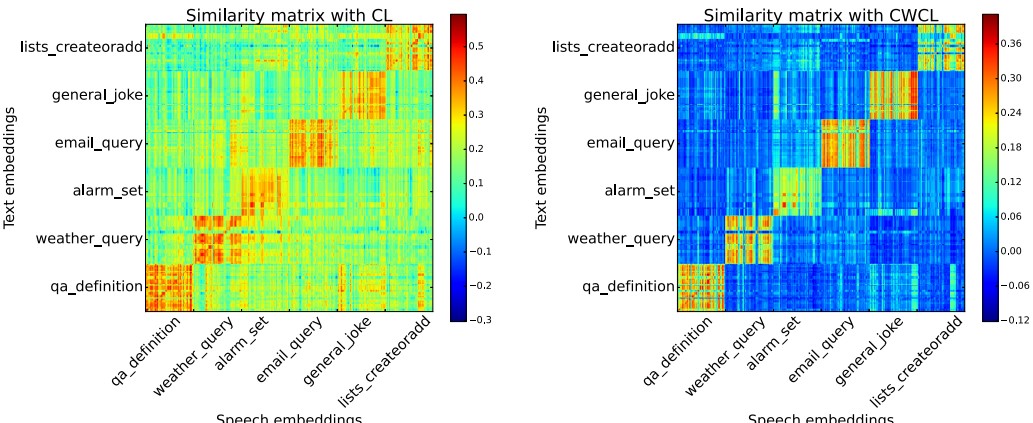

Figure 8: Another example of alignment between speech and text embeddings. The sampled classes are different from those used in Figures 3 and 7.

### A.3.4 Additional tables for reference

Table 8 additionally shows the Top-5 accuracy over speech-text experiments. Since most of the previous works did not report this metric, we only include our own experimental results. Table 9 shows the existing supervised model performances where the models are either trained or fine-tuned on the labeled Google Speech Command Dataset V2 for performing the keyword spotting (KWS) task, while our methods did not require any labeled KWS data for performing the task.

### A.3.5 General template as a python list

To test the speech-text alignment models, we use a "general" set of templates in addition to the one obtained by using the text from the training data itself. This general set of templates aims to mimic a scenario where the no example texts maybe available. We list the set of template sentences used in the general set here.

Table 8: Top-5 accuracy for zero-shot speech-to-intent classification (SLURP and STOP) and KWS on Google Speech Command Dataset V2. Superscript # is used to indicate use of general templates.

| Method | Text model | SLURP | SLURP# | STOP | STOP# | KWS | KWS# |
|---|---|---|---|---|---|---|---|
| CL | RoBERTa+S | 69.57 | 49.86 | 98.19 | 94.03 | 82.22 | 82.53 |
| CL | BART+Y | 52.97 | 24.87 | 95.27 | 81.63 | 84.02 | 78.14 |
| **CWCL (Ours)** | RoBERTa+S | 84.53 | 68.58 | 99.38 | 96.52 | 91.20 | 92.42 |
| **CWCL (Ours)** | BART+Y | 79.48 | 57.34 | 99.48 | 97.71 | 93.79 | 94.30 |
| Text-intent | RoBERTa+S | 95.66 | 83.36 | 98.93 | 95.20 | 100 | 98.20 |
| (upper bound) | BART+Y | 99.58 | 73.82 | 99.45 | 98.40 | 100 | 100 |

Table 9: Keyword spotting Top-1 accuracies on Google Speech Command Dataset V2 from existing supervised models.

| Method | KWS |
|---|---|
| Attention RNN [47] | 93.9 |
| KWT-2 [41] | 97.74 |
| Wav2Vec2 [48] | 96.6 |
| M2D [49] | 95.4 |
| M2D - Fine tuned [49] | 98.5 |

General template sentences: [ it is about { }, it was about { }, it will be about { }, this is about { }, this was about { }, this will be about { }, it is related to { }, it was related to { }, it will be related to { }, this is related to { }, this was related to { }, this will be related to { }, it is talking about { }, it was talking about { }, it will be talking about { }, this is talking about { } this was talking about { }, this will be talking about { }, I am talking about { }, I was talking about { }, I will be talking about { }, You are talking about { }, You were talking about { }, You will be talking about { }, They are talking about { }, They were talking about { }, They will be talking about { }, We are talking about { }, We were talking about { }, We will be talking about { }, it talks about { }, it talked about { }, it will talk about { }, this talks about { }, this talked about { }, this will talk about { }, I talk about { }, I talked about { }, I will talk about { }, You talk about { }, You talked about { }, You will talk about { }, They talk about { }, They talked about { }, They will talk about { }, We talk about { }, We talked about { }, We will talk about { } ]