# OpenReview forum: "CWCL: Cross-Modal Transfer with Continuously Weighted Contrastive Loss"
_NeurIPS.cc/2023/Conference — NeurIPS 2023 poster_

### Official Review · Reviewer_1ozP · 2023-07-05

**Soundness:** 3 good
**Presentation:** 2 fair
**Contribution:** 3 good
**Rating:** 6
**Confidence:** 4

**Summary:**

The paper proposes a novel continuously weighted contrastive loss function for multi-modal models. The proposed method takes consideration of 1) similarity of samples in the same batch 2) the non-binary nature of similarity between input samples,  and therefore utilizes a continuous weight matrix in learning alignment between modalities. In general I think that the ideas on this paper are novel. The explanation of the methods is explicit and concrete. I think it is enough to be reproducible. However the paper may benefit from some revision regard to how the work is presented and focus of experiments.

**Strengths:**

* The proposed idea is novel in resolving two crucial problems in standard contrastive loss.
* The authors considered modalities of both text-image and speech-image and presented some supportive zero-shot evaluation results.

**Weaknesses:**

* The experimental results are not complete, it would be good to see a more complete set of results on how this method would perform on the downstream retrieval tasks (T4 compares between CL and CWCL, without indicating performance compared to baseline models such as LiT or CLIP).
* The experiments regarding the transfer from text to speech are nice however the results in Section 4.2.1 is not clearly presented and is a bit hard to follow.
* Figure1 is not really informative and maybe can be combined with Figure 4 so that more interesting  and qualitative results can be presented.


**Questions:**

* Is it possible for you to provide results for retrieval compared to the baseline models? This would address a major weakness in the evaluation.

**Limitations:**

* There is no explicit limitations section in the paper.

---

> ### Author Rebuttal · Authors · 2023-08-10
>
> **W1: Results are incomplete...**
>
> Thank you for raising this point. In Table 4, the row corresponding to CL is indeed the LiT model. We named it CL to indicate that this model was tuned using standard contrastive loss as done in the LiT paper. In fact, the model referred to in this row follows the same training procedure as LiT, with the image encoder initialized using a pre-trained model and frozen during training. We will make this change and rename the row to LiT to clarify this.
>
> Further, the results reported in Table 4 correspond to using the ViT-L/16 architecture for the image encoder and a 12-layer transformer model for the text encoder; trained using the YFCC15M and the CC12M datasets. However, we noticed that the LiT model achieves the best performance (using a publicly available dataset) when the BERT-large model is used as the text encoder.
>
> Thus, we performed further experiments with the BERT-large model as the text encoder. We again observe an improvement in the retrieval performance. We provide the results in Table 12. Further, we also observe performance gains in the zero-shot image classification task as well. We provide the results in Table 13.
>
> In the paper, we did not compare CWCL against CLIP on the retrieval task, since we expect CLIP to have lower performance than LiT (and CWCL already performs better than LiT). However, we realize that a direct comparison between CWCL and CLIP might be of interest. To this end, we provide a direct comparison of the two methods using zero-shot retrieval performance on the MS-COCO dataset. Due to the limited time frame, we trained new CWCL and CLIP models on a subset of the training data that consists of samples from/ only CC12M. Further, we trained both models for only 20 epochs. Although this setup results in low absolute performance on both models, we can still obtain a direct relative comparison. We outline the results in Table 14. Interestingly, the models trained on the CC12M dataset (Table 14) perform better at retrieval than models trained on both YFCC15M and CC12M (Table 4). We have noticed such behavior in open-source CLIP models (\url{https://github.com/mlfoundations/open_clip}) as well. We hypothesize that this is because CC12M is a dataset that has been subject to filtering out bad samples, as opposed to YFCC15M which is not.
>
>
> We hope that we have addressed the reviewer's concerns about the results of the retrieval task performance of CWCL based approach. We are happy to answer any further questions.
>
>
> **W2: Exp Text to Speech...**
>
> Thank you for this comment. We reviewed and edited the entire section 4.2 carefully considering readers could be less familiar than us with the contents we wrote. We were also short on space and hence combined all results into a single table (Table 2 in the original manuscript).
> In the revised paper, we will improve this section for readability.
>
> **W3: Figure 1...**
>
> Thank you for this suggestion. We will combine Figures 1 and 4.
>
> **Q1: Is it...**
>
> We have provided the results of retrieval tasks in our response to one of your earlier comments. Further, we also provide results based on new experiments that studied the ViT-L-16+BERT-large configuration. We appreciate the reviewer's feedback regarding our experiments and we hope we have addressed their concern.
>
> **Summary**
>
> Overall, we appreciate your questions and suggestions. They have helped us improve the paper. We agree that including performance on retrieval tasks adds to the completeness of our results. We will include these in the main paper manuscript. We will also clarify the results in Section 4.2.1. We were a little pressed for space in the original manuscript but we will better manage the space in the revised version by moving Figure 1. This will help us clarify Section 4.2.1 further.
>
> We hope that we have addressed your concerns about our paper and sincerely hope to discuss further with you during the discussion phase.

---

> > ### Comment · Reviewer_1ozP · 2023-08-15
> >
> > Thanks to the authors for their detailed response. You have addressed my concerns so I will increase my rating from 5 to 6.

---

> > > ### Author Response · Authors · 2023-08-15
> > >
> > > We thank the reviewer for their prompt response and consideration!

---

### Official Review · Reviewer_q9nH · 2023-07-06

**Soundness:** 3 good
**Presentation:** 3 good
**Contribution:** 2 fair
**Rating:** 4
**Confidence:** 3

**Summary:**

The authors propose a composite loss in contrastive learning in order to preserve structure between two embedding spaces for different modalities. In the setting in which they study, two unimodal encoders are aligned by freezing one of them and computing a "continuous" contrastive weight between samples $(i, j)$ as $0.5 + \frac{q_i^T q_j}{2}$. With these weights an additional loss for structure preservation is used in addition to the standard self-supervised loss for the unfrozen encoder. The proposed approach outperforms standard approaches such as LiT, SimCLR on 0 shot classification tasks.

**Strengths:**

- The method is simple and efficient requiring a small $\mathcal(O)(k^2)$ computation for pairwise weights per batch
- Performance improvements are strong across the experimentation

**Weaknesses:**

- There is a lot of setup while the entirety of novel contributions seem to be buried in section 2.3.
- This paper makes more sense for a workshop, the contributions are limited and is a small modification of previous works such as (https://arxiv.org/pdf/1905.12837.pdf, https://arxiv.org/pdf/1811.01459.pdf, https://arxiv.org/pdf/2103.14003.pdf)

**Questions:**

- Which continuous pairwise weights did the authors try? Given this is the main contribution it would be helpful to understand performance as this weighting changes in the 0.5 bias used or for nonlinear weighting wrt to the inner products.

---

> ### Author Rebuttal · Authors · 2023-08-10
>
> **W1: There is a...**
>
> Please note the problem setup is only done in Section 2.1. We request the reviewer to elaborate further on what they mean by "lot of setup".
>
> Further, our contributions are multi-fold and are not confined to Section 2.3. The novel CWCL loss function is introduced in Section 2.2, while Section 2.3 discusses how we calculate the similarity weights.
>
> Additionally, we demonstrate the effectiveness of the proposed methods using many experiments, models, datasets, and modalities. For example, we show the "template robustness" property of the proposed methods and we provide qualitative results showing that the proposed loss function leads to better alignment between the modalities.
>
> We believe that this is the first work to study 0-shot speech-intent classification where no task-specific speech data was used for training. Further, our results show a strong improvement over methods that use standard contrastive loss. We would consider these results to be novel.
>
> In general, we request the reviewer to elaborate on why they feel that these other aspects are not novel. We believe that in a field like machine learning, an idea followed by extensive experimental results that demonstrate the effectiveness of the idea together contribute to novelty.
>
>
> **W2: This paper...**
>
> *The General Pair-based Weighting Loss for Deep Metric Learning (\url{https://arxiv.org/pdf/1905.12837.pdf})*:
>
> The referred paper is very different from ours for the following reasons:
>
> Firstly, it studies the problem of metric learning in a supervised setting, where class information is available in the dataset. This is mentioned explicitly in Section 2.A of the paper where they formulate the problem and in Algorithm 1(authors mine “positive” and “negative” examples using Equations (16, 17) by using class labels).
>
>
> Secondly, they do not consider multi-modality. The algorithms proposed in this paper are geared towards a single modality. Hence, the resulting models cannot be used for other downstream tasks in a zero-shot way. On the contrary, they can only be used for the specific task that they are trained on.
>
> In our paper, we have already acknowledged a weighted version of contrastive loss has been developed in the supervised learning case by Khosla et. al., (NeurIPS 2020, ref. [14] in our paper). However, extending weighted contrastive loss to self-supervised learning has not been studied before. Further, models such as CLIP and LiT have proven that the scale of data and models have a large impact on downstream performance. But they only use the standard contrastive loss because developing a weighted contrastive loss in the context of the work was not explored before. Thus, our work is not a simple modification of the above paper.
>
> *Deep Metric Learning by Online Soft Mining and Class-Aware Attention\url{https://arxiv.org/pdf/1811.01459.pdf}*:
>
> This also addresses the problem of metric learning in a supervised setting. This is highlighted in the main methodology section, where they consider a uni-modal dataset of the form ${x_i, y_i}$ where $y_i$ is the class label. This class label is used to define the positive set $P$ (samples with the same class label) and negative set $N$ (samples with different class labels), used to compute the losses in eqns. (7, 8, 9).
>
> *Rethinking Deep Contrastive Learning with Embedding Memory\url{https://arxiv.org/pdf/2103.14003.pdf}*:
>
> This also addresses the supervised learning case; as we quote the introduction section of the above paper: “In this work, we simply adopt MoCo in a supervised manner for DML, which is referred to as s-MoCo. It provides a strong example for us to study the pair weighting in memory-based DML”.
>
> **Q3: Which..**
>     Please refer to the answer we provided to Reviewer 2's questions on the different similarity functions that we considered.
>
>
> **Overall comments on comparison with the above three papers:**
>
> We understand the reviewer’s concern that these papers study variations of weighted contrastive loss. However, extending weighted contrastive loss from a supervised setting to a self-supervised setting is non-trivial. To summarize the difficulty, all three papers pointed out consider the following question: Using prior information about positive and negative pairs, how to weigh positive pairs that are further apart to bring them closer, and weigh negative pairs that are closer so that they are pushed apart? This is captured in Figure 2. of the third paper pointed out by the reviewer (https://arxiv.org/pdf/2103.14003.pdf). Thus, an optimal solution here is where all positive samples are brought as close as possible, and all negative samples are pushed away as far as possible. On the contrary, in our work, there is no notion of strictly positive or negative. This is a core contribution of the paper, a softer notion of similarity helps us recreate the embedding space in one modality based on a pre-trained model from another modality.
>
> The difference between bringing together all positive samples and repelling all negative samples (with weighted or unweighted losses) and that of considering similarity as a continuous metric and learning a continuous structure in the embedding space of a modality may be subtle, but the resulting algorithms and models have highly different capabilities and applications. Using supervised labels to construct positive and negative sets makes the models specific to those class labels; features that distinguish two inputs within a class are filtered out and features that distinguish two inputs from different classes are accentuated. However, since our models do not make use of class-specific labels, they retain all features and are capable of zero-shot classification on many downstream tasks.
>
> We hope we have addressed the reviewer’s concerns regarding the relationship between weighted contrastive loss in a supervised setting and our work and invite them for more discussion.

---

### Official Review · Reviewer_yjNF · 2023-07-11

**Soundness:** 3 good
**Presentation:** 3 good
**Contribution:** 2 fair
**Rating:** 6
**Confidence:** 4

**Summary:**

In this work, the authors consider the problem of cross-modal zero-shot transfer and propose a new loss function called continuously weighted contrastive learning (CWCL) that extracts better supervision from pretrained models in a single modality and leads to better alignment between two modalities. They run experiments on two modality pairs, image-text and speech-text, and CWCL yields significant improvements compared to standard contrastive learning (with up to 20-30% absolute improvement on a speech-to-intent classification task).

**Strengths:**

- The authors present a simple and effective contrastive learning variant that makes effective use of pretrained models in individual modalities.
- They achieve a significant performance boost in zero-shot image classification when compared to existing contrastive learning objectives.
- They also demonstrate superior results on a zero-shot speech-to-intent classification task.

**Weaknesses:**

The main contribution of this work (CWCL) can be grounded better in the context of existing work. This work highlights an important shortcoming in existing formulations of contrastive learning, namely that similarity has a binary interpretation. All positive examples and all negative examples are treated equally. This has been identified in prior work, albeit for a single modality, which the authors fail to cite. E.g., "Not All Negatives are Equal: Label-Aware Contrastive Loss for Fine-grained Text Classification", V. Suresh and D. Ong, EMNLP 2021. There are also works on multimodal learning that highlight the importance of using a weighted contrastive loss to downweight faulty positives/negatives such as "Robust Audio-Visual Instance Discrimination", Morgado et al., CVPR 2021.

**Questions:**

- Did the authors experiment with other weighting functions for w_ij's in Equation 3? Some insights on how w_ij = q_i^T q_j/2 + 0.5 was arrived at would be useful for the reader.
- As in Figure 5, does CWCL exhibit robustness to templates for the speech-to-intent classification task as well?
- From Table 1, CWCL outperforms the supervised Resnet50 baseline on ImageNet-V2 as well as ImageNet-R, ImageNet-A and ObjNet by substantially larger margins. Is Resnet50 the appropriate supervised baseline to show for these datasets?
- From Table 2:
    - RoBERTa+S performs consistently better than BART+Y on SLURP and STOP, but the trend flips for KWS. Can the authors comment on why that is?
    - CWCL with RoBERTa+S appears to outperform the text-intent upper bound for STOP (87.87 vs. 84.78). Please explain this result.
- Just a comment: If the authors are looking to make some space in the draft, Figure 1 can be omitted without loss of clarity.

**Limitations:**

Limitations have not been explicitly listed.

---

> ### Author Rebuttal · Authors · 2023-08-10
>
> **W1 : The main...**
>
> Both references are very interesting and relevant and we will cite them.  We provide a detailed comparison below.
>
> **Comparison with "Not all Negatives are Equal:.."**
>      The ideas explored in this paper are similar in spirit to those considered in our paper. However, this paper considers supervised learning. In our work, we consider fully self-supervised learning. This lets us use our models on many downstream tasks in a zero-shot way, whereas in the supervised setting, the resulting models are applicable only to the specific task that they are trained for.
>
> **Comparison with "Robust Audio-Visual Instance Discrimination" (soft-XID)}:**
>
> It is our belief that when models are trained initially using standard contrastive loss and then trained further by using weighted contrastive loss, biases from the initial training stage get further reinforced later on. In the initial stage of training, the encoders are trained to repel all samples except only samples belonging to the same pair. When the same encoders are used to obtain the similarity scores in the second stage, these similarity scores are unreliable and will further reinforce the biases from the first stage.
>
> We performed an experiment that similar in spirit to the above paper. We trained a pair of image and text encoders using the standard contrastive loss using the CC12M dataset for 20 epochs. Note that this is similar to the setup of the CLIP model. After this, we fine-tuned the model using weighted contrastive loss for another 20 epochs. In the second stage, we obtain the similarity scores using the encoders trained in the first stage, as proposed in "Robust ..." paper. We then measure the zero-shot image classification accuracy on ImageNet. We also train a model using the proposed CWCL method for a total of 40 epochs. We provide the results below in Table 11. We further note that in the first experiment (emulating soft-XID method) the 0-shot accuracy decreased upon fine-tuning, further indicating a conflict between the similarity scores generated using the models trained using standard contrastive loss.
>
> Overall, we find that the method of first using standard contrastive loss and then using a weighted contrastive loss yields worse performance as compared to our proposed method.
>
> **Q1 Did...**
>
> We first considered a softmax-based weighting function defined as over the intra-modal similarities and secondly our proposed function.      The second formulation performs slightly better than the first one. Further, it results in a nicer interpretation of the proposed CWCL loss function as an interpolation between standard contrastive loss and the supervised contrastive loss function, as mentioned in lines 130-134 in the main paper. Further, note that in the second formulation, embeddings with $\langle q_i, q_j\rangle = -1$ result in a weight of 0, thus not resulting in any "attractive force" between such samples.
>
>
> **Q2: As in...** Yes, we provide thein Table 10 of the rebuttal PDF.
>
> **Q3: From Table 1...**
>
> One of the advantages of models such as CLIP, LiT, and CWCL is that the same model can be used on multiple downstream tasks and datasets in a zero-shot way. To perform a more direct comparison to this regime, we choose a supervised and fine-tuned model that achieves a comparable performance on ImageNet and we report the same model's (after fine-tuning) accuracies on the other datasets. We wanted to better demonstrate how using a single model for multiple tasks performs and compare it with the CWCL-based model. Further, we also wish to note that the authors in Zhai et.al.,(LiT) make the same comparison.
>
> **Q4: From Table 2 RoBERTA+S...**
>
> The pre-trained text model used in RoBERTa+S was trained on the data including SLURP intent classification data (but only the text data. The SLURP speech data was not used in any stage of the training. Further, no data from STOP was used in the pre-training stage). We think that this affected the model to overfit on the task of intent classification in general, outperforming BART+Y (based on performances on SLURP and STOP) while performing worse in keyword spotting (KWS) than BART+Y.
>
> **Q5: From Table 2 CWCL...**
>
> We would like to clarify that in using the term "upper bound", we meant to convey that the text models' performance on intent classification serves as a "reference" for what can be expected from the speech encoder after training. We elaborate further below.
>
> In our method, we "freeze" the text encoder and train the speech encoder using a paired (speech-text) dataset. In particular, we use the commonVoice dataset. Note that the text model does not get updated and hence does not get a chance to process this dataset. Further, the speech encoder was initialized using a pre-trained ASR model's encoder. A third difference between the text model and the speech mode is that the speech tower might have learned how to utilize acoustic cues in addition to linguistic information from a given speech utterance, to align its embedding to the semantic embedding from the text tower.  These three differences could contribute to the speech model sometimes performing better than the text model. However, we still expect the text model's performance to be a good reference for what the speech model can achieve after training.
>
> Another factor that might have led to this is that the number of classes in the STOP dataset is low (8), as opposed to the SLURP dataset (which has 60 classes). Thus, in the case of SLURP, the performance on the speech-intent task is lower than the text-intent task, since it is more challenging.

---

> > ### Comment · Reviewer_yjNF · 2023-08-19
> > **Response to rebuttal**
> >
> > Thanks to the authors for their detailed response and clarifications to my specific questions. I'm raising my score from 5 to 6.

---

> > > ### Author Response · Authors · 2023-08-19
> > >
> > > We thank the reviewer for prompt response and consideration! We appreciate your comments and feedback on our paper.

---

### Official Review · Reviewer_5jbW · 2023-07-13

**Soundness:** 3 good
**Presentation:** 4 excellent
**Contribution:** 3 good
**Rating:** 7
**Confidence:** 3

**Summary:**

They propose a simple but effective method to align the representation space of two self-supervised models using pairs of examples from two modalities. They propose a CWCL loss where they reweight the contrastive loss of example pairs based on the similarity measured in one modality (equation 2). Specifically, let $(v_1, u_1)$ and $(v_2, u_2)$ be two example pairs, the CWCL loss encourage the representation of $v_1$ to be close to $v_2$ if $u_1$ is close to $u_2$. They experiment with a task requiring text-image and a task requiring text-audio alignment in zero-shot settings. The results show that adding CWCL loss to the original contrastive loss outperforms previous methods greatly.


**Strengths:**

1. The proposed approach is very simple, but the results show that it brings significant improvements. Especially for the image classification task, their model outperforms the supervised-trained model in the zero-shot setting.
2. They also show that, compared to a previous method, LIT, their model is more robust to the choice of prompt templates used to generate the text representation for the zero-shot image-classification experiment.
3. The experiment of audio-text alignment also shows that adding the CWCL loss improves the performance.


**Weaknesses:**

1. The ablation study comparing pure contrastive loss vs contrastive loss plus CWCL is only conducted for the audio-text alignment task.
2. In section 4.2.1, the author selected the text model very carefully. It thus seems that this method is very sensitive to the choice of the text model.
3. It may provide more insight if there are some qualitative analyses between the model trained without CWCL and the model trained with CWCL.


**Questions:**

1. The performance of LiT reported in Table 1 is much lower than the performance reported in the LiT paper. Could you explain what might be the reason for this discrepancy?
2. For the zero-shot keyword spotting task, because this task seems to be very much at the lexical level instead of the semantic level, I wonder what would be the benefit of using a contextualized pretrained language model?
3. Instead of having CWCL loss in addition to the original contrastive loss, what about having CWCL loss only but you set the weight of negative pairs to be 0? There seem to be some subtle design choices made for CWCL. I would like to know more.

**Limitations:**

I didn't find the authors address the limitations of this work.

---

> ### Author Rebuttal · Authors · 2023-08-10
>
> **W1  The ablation study...**
>
> We provide the comparison between standard contrastive loss and CWCL for the image-text modality in Table. 1. Please note that we have chosen to name the standard contrastive loss-based training as LiT, since we follow their training protocol by first initializing with a pre-trained image model and then using the standard contrastive loss as the training objective.
>
> The same comparison for the speech-text modality pair is provided in Table. 2. For standard contrastive loss, we initialize both speech and text encoders with pre-trained models, freeze them, and train the speech encoder only. In this regard, it is similar to LiT. However, we name this method as CL and not LiT, since LiT does not consider this pair of modalities, as its name suggests **L**ocked-**I**mage **T**uning, and we wanted to avoid any misrepresentation of the LiT paper.
>
> Hence, the ablation studies presented in Table 1 and 2 are equivalent and consider the image-text and speech-text modalities respectively. We will clarify this further in the revised paper.
>
> **W2 In section 4.2...**
>
> We agree that the text model choice could affect the final model performance in the downstream tasks. However, when selecting the two text models used in this paper, we simply selected 2 models that performed the best in the text intent classification on SLURP, among 50 publicly available text models (the details are written in 4.2's model architecture section). Please note the CWCL performance can be expected to be better than CL performance irrespective of the choice of the text model. We chose those text models to demonstrate good performance on downstream tasks. All that said, how to choose the right models for the training in detail is very much worth exploring in a separate work and we appreciate this comment.
>
> **W3: It may...**
>
> In Figures 3, 7, and 8 in the original manuscript, we provide examples of the alignment between the audio and the text embeddings generated using the SLURP speech-to-intent classification dataset. Speech (audio) data in this dataset was never exposed to the model at any stage of training. In this figure, we can observe that using CWCL leads to a strong alignment between speech samples and text samples that have similar semantic meanings. For example, all the speech samples related to queries about the news have a high alignment with the text samples that have similar meanings. (We would like to emphasize that this alignment is observed in a downstream dataset whose speech data was not used during training.) However, we do not observe such an alignment in the models trained with only standard contrastive loss (CL), as seen in the figures. Hence, the speech embedding model trained with CWCL can be considered to have a better language understanding than one trained with CL.
>
> A second set of qualitative results are provided in Section 4.1.2, where we study the robustness to templates in the image-text model. Owing to a stronger alignment between images and text, models trained using CWCL are able to generalize better under variations in the template sentences used.
>
> **Q1: The perf...**
>
> This is a good question! It is because of the architecture we used while training the models. In our experiments, we used the ViT-L/16 architecture for the image encoder and a 12-layer transformer for the text encoder. However, LiT achieves its best performance when the BERT-large architecture is used for the text encoder.
>
>  In order to address this, we have now trained models using the ViT-L/16 + BERT-large configuration. We followed the training procedure described in the LiT paper as closely as possible. We provide results with the updated model architecture in Table 13 (the PDF of the global rebuttal). We are able to achieve much higher performance with the new architecture (LiT: 71.2%, CWCL: 76.48%).
>
> **Q2: For the...**
>
> We agree that the task of keyword spotting is at a lexical level than a semantic level. However, our goal is to train a versatile model that can be used for multiple tasks, e.g., speech-to-intent classification and keyword spotting.
>
> While using our model for keyword spotting, we convert the keywords to sentences and also convert the keyword classification task to the alignment task. This is similar to 0-shot image classification done by CLIP and LiT-like models, which is also a task that does not require contextual language information. Further, the contextualized pre-trained language model could learn richer information in words considering the context in which each word is used. So our model performs well even when the keywords are used as is (without converting them to sentences).
>
> Although we followed LiT's training protocol as closely as possible, the LiT model we trained achieves a zero-shot accuracy of 71.2\% on ImageNet (as opposed to 75.7\% mentioned in the LiT paper). We hypothesize that this difference is due to some training hyperparameters not fully specified in the LiT paper (such as the details of the Adafactor optimizer) and some possible differences in the training dataset (since some of the URLs in the CC12 and YFCC datasets may have become inactive, hence affecting our download of the dataset). However, under the same training setup, the CWCL model achieves 76.48\%.  In general, we again find that the CWCL-based models achieve much better performance.
>
> **Q3: Instead...**
>
> This is an interesting suggestion. When the weights for the negative pairs are set to 0, the resulting loss function is close to the standard loss function, except that the weights for the positive pairs can still be tuned. This might help with dealing with "faulty positives". We thank the reviewer for their suggestion.
>
>  In the current paper, we do not explore this and simply set the weights for the positive pairs to be 1, which results in the standard contrastive loss function.

---

> > ### Comment · Reviewer_5jbW · 2023-08-10
> >
> > Thank for the clarifications. I don't have a follow-up question at this moment. I would suggest the authors include these clarifications in their revision.

---

> > > ### Author Response · Authors · 2023-08-10
> > >
> > > We thank the reviewer for their quick and prompt response to our rebuttal. We will make sure to include the above clarifications in the revised paper.

---

### Author Rebuttal · Authors · 2023-08-10

We would like to thank all the reviewers for their time and effort in reviewing our paper. We appreciate all of the comments and they have helped us improve our paper. Here, we first provide a summary of our paper. Then, we outline the major concerns expressed by the reviewers and explain how we have addressed them. Following this, we also provide a summary of the positive comments made by the reviewers.

**Paper summary:**
 1. We consider the problem of cross-modal transfer, where representations in one modality are learned using a pre-trained model in another modality and a paired dataset that consists of pairs of the two modalities.
2. To address inefficiencies in the training objective used in existing work, we propose a novel loss function called "Continuously Weighted Contrastive Loss" (CWCL) that considers associations between all the pairs in a training batch.
3. We show that the new loss function leads to a 5-8% (absolute) increase in 0-shot task performance for the image-text modality pair. We also study our method in the context of speech-text modality pair. We provide the first results in 0-shot speech-to-intent classification and keyword spotting, where our models achieve 20-30% (absolute) improvement over existing methods.

Overall, the reviewers had the following positive comments about our paper: The reviewers feel that our work addresses an important shortcoming in existing formulations (yjNF, 1ozP)and that we propose a novel solution that is also simple and efficient (q9nH). All the reviewers expressed that our experimental results are very strong. Reviewers also found the template-robustness property of our models interesting (5jbW).

**Summary of reviewer concerns**
We categorized the reviewers' concerns into broader themes. We describe them below by providing a summary of our responses to each of them.

**Completeness of experiments and interpretation of the results:** Reviewer 5jbW felt that some experiments were missing for the image-text modality pair, yjNF had some questions about experimental results, reviewer q9nH had a question about the design choice for the similarity function, and 1ozP asked for more results on the retrieval task for the image-text modality pair. We summarize our responses below:
 - We provide new experimental results in the rebuttal including a direct comparison between our method, CLIP, and LiT on the retrieval task. We also provide updated results for all the image-text experiments with a new model architecture based on the ViT-L-16+BERT-large configuration.
- In particular, our new image-text model achieves a 0-shot ImageNet accuracy of 76.5\%, which is better than the best baseline (LiT) when trained on the same dataset.
- We have provided explanations for the choice of our baseline models in our response to reviewer yjNF. We have also provided explanations for some of the results in Table 2 regarding the 0-shot performance in speech-text tasks.
- We have addressed why we chose the similarity function that we use in our paper.
- We realized that we used two different naming conventions (`CL` and `LiT`) for the baseline method that uses the standard contrastive learning loss. This led to some confusion regarding the equivalence of the experiments between the image-text and speech-text modality pairs. We will correct this to ensure uniformity.

**Comparison to existing work:** Reviewers yjNF and q9nH pointed us to a few papers that are relevant to our work. We provided details responses comparing our work to them in the rebuttal and we summarize it here.

All but one of the papers pointed out by the reviewers consider the weighted contrastive loss for supervised learning, where class information is available. We have cited the work of Khosla et.al., in our main paper which also considers contrastive loss in the supervised case. In our work, we consider self-supervised learning. In the former case, conditioned on
prior information about which pairs are positive pairs and which are negative pairs, positive pairs that are further apart are weighted highly to bring them closer, and negative pairs that are closer are weighted highly so that they are pushed apart strongly. On the contrary, in our work, there is no notion of strictly positive or negative. This is a core contribution of the paper since a softer notion of similarity helps us recreate the embedding space in one modality based on a pre-trained model from another modality.

One of the papers considers self-supervised learning and is more similar to our paper. However, since they do not use pre-trained models, they need to use a "self-teaching" process, where a model is trained initially using only standard contrastive loss and then fine-tuned using a weighted contrastive loss.
During the rebuttal phase, we performed experiments with the self-teaching strategy and we found that that performance usually does not improve much or even degrades in our experiments. This may be due to the conflicting nature of the two stages of training. We provide experimental results in our response. Further, The weighted loss they propose is designed to "down-weight" the effect of faulty positives and "up-weight" faulty negatives. In our method, we do not consider samples as positives or negatives.

**Presentation of the paper:** We also received some comments about the presentation in our paper. We will address these concerns in the revised version of our paper.

---

### Decision · Program_Chairs · 2023-09-21

**Decision:**

Accept (poster)

**Comment:**

This paper proposes a contrastive objective for cross modal transfer. I think the idea is interesting and the experiments empirically validate its efficacy. Reviewer q9nH has some concerns around the novelty of the proposed method. However, I think the authors have sufficiently addressed this in their rebuttal. All other reviewers provided positive feedback for the work. I recommend accepting the paper to the conference.